# Drill-down: Interactive Retrieval of Complex Scenes using Natural Language Queries

**Fuwen Tan**
University of Virginia
fuwen.tan@virginia.edu

**Paola Cascante-Bonilla**
University of Virginia
pc9za@virginia.com

**Xiaoxiao Guo**
IBM Research AI
xiaoxiao.guo@ibm.com

**Hui Wu**
IBM Research AI
wuhu@us.ibm.com

**Song Feng**
IBM Research AI
sfeng@us.ibm.com

**Vicente Ordonez**
University of Virginia
vicente@virginia.edu

## Abstract

This paper explores the task of interactive image retrieval using natural language queries, where a user progressively provides input queries to refine a set of retrieval results. Moreover, our work explores this problem in the context of complex image scenes containing multiple objects. We propose Drill-down, an effective framework for encoding multiple queries with an efficient compact state representation that significantly extends current methods for single-round image retrieval. We show that using multiple rounds of natural language queries as input can be surprisingly effective to find arbitrarily specific images of complex scenes. Furthermore, we find that existing image datasets with textual captions can provide a surprisingly effective form of weak supervision for this task. We compare our method with existing sequential encoding and embedding networks, demonstrating superior performance on two proposed benchmarks: automatic image retrieval on a simulated scenario that uses region captions as queries, and interactive image retrieval using real queries from human evaluators.

## 1   Introduction

Retrieving images from text-based queries has been an active area of research that requires some level of visual and textual understanding. Significant improvement has been achieved over the past years with advances in representation learning but finding very specific images with detailed specifications remains challenging. A common way of specification is through natural language queries, where a user inputs a description of the image and obtains a set of results. We focus on a common scenario where a user is trying to find an exact image, or similarly where the user has a very specific idea of a target image, or is deciding on-the-fly while querying. We present empirical evidence that users are much more successful if they are allowed to refine their search results with subsequent textual queries. Users might start with a general query about the "concept" of the image they have in mind and then "drill down" onto more specific descriptions of objects or attributes in the image to refine the results.

Among previous efforts in image retrieval, a promising paradigm is to learn a visual-semantic embedding by minimizing the distance between a target image and an input textual query using a joint feature space. Pioneering approaches such as [17, 34, 9, 21, 36, 33] have demonstrated remarkable

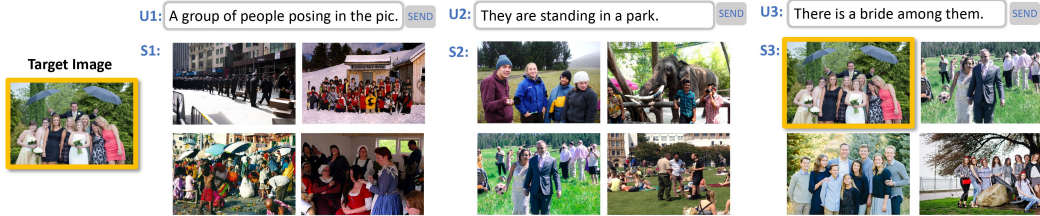

Figure 1: An example of the interactive image retrieval with our Drill-down model, where a user generated query (U$t$) progressively refines the search results (S$t$) until the target image is among top search results.

performance on large scale datasets such as Flickr30K [26] and COCO [23], and domain-specific tasks such as outfit composition [12]. However, we find that these methods are limited in their capacity for retrieving highly specific images, because it is either difficult for users to be specific enough with a single query or users may not have the full picture in mind beforehand. We show an example of this type of interaction in Figure 1. While single-query retrieval might be more suited for domains such as product search where images typically contain only one object, requiring users to describe a whole scene in one sentence might be too demanding. More recently, dialog based search has been proposed to overcome some of the limitations of single-query retrieval [22, 31, 10, 7].

In this paper, we propose Drill-down, an interactive image search framework for retrieving complex scenes, which learns to capture the fine-grained alignments between images and multiple text queries. Our work is inspired by the observations that: (1) user queries at each turn may not exhaustively describe all the details of the target image, but focus on some local regions, which provide a natural decomposition of the whole scene. Therefore, we explicitly represent images as a list of object/stuff level features extracted from a pre-trained object detector [27]. This is also in line with recent research [21, 36] on learning region-phrase alignments for single-query methods; (2) complex scenes contain multiple objects that might share the same feature subspace. Particularly, existing state representations of sequential text queries, such as the hidden states of a RNN, condense all image properties in a single state vector, which makes it difficult to distinguish entities sharing the same feature subspace, such as multiple `person` instances. To address this, we propose to maintain a set of state vectors, encouraging each of the vectors to encode text queries corresponding to a distinct image region. Figure 2 shows an overview of our approach, images are represented with local feature representations, and the query state is represented by a fixed set of vectors that are selectively updated with each subsequent query.

We demonstrate the effectiveness of our approach on the Visual Genome dataset [20] in two scenarios: automatic image retrieval using region captions as queries, and interactive image retrieval with real queries from human evaluators. In both cases, our experimental results show that the proposed model outperforms existing methods, such as a hierarchical recurrent encoder model [29], while using less computational budget.

Our main contributions can be summarized as follows: [1]

- We propose Drill-down, an interactive image search approach with multiple round queries which leverages region captions as a form of weak supervision during training.
- We conduct experiments on a large-scale natural image dataset: Visual Genome [20], and demonstrate superior performance of our model on both simulated and real user queries;
- We show that our model, while producing a compact representation, outperforms competing baseline methods by a significant margin.

## 2   Related Work

Text-based image retrieval has been an active research topic for decades [5, 4, 28]. Prominent more contemporary works have recognized the need for richer user interactions in order to obtain higher

quality results [30, 18, 19, 2]. Siddiquie et al [30] proposed an approach to use multiple query attributes. Kovashka et al [18, 19] further proposed using user feedback based on individual visual attributes to progressively improve search results. Arandjelovic et al [2] proposed a multiple query retrieval system that was used for querying specific objects within a large set of images. These works show that multiple independent queries generally outperform methods that jointly model the input set with a single query. Our work builds on these previous ideas but does not use an explicit notion of attributes and aims to support more general input text queries.

Remarkable results have been achieved by recent methods based on deep learning [17, 34, 9]. These methods typically explore mapping a text query and the target image into a common feature space. Learned feature representations are designated to capture both visual and semantic information in the same embedding space. In contrast, besides supporting multiple rounds of queries, our approach also has a richer region representation to explicitly map individual entities in images to textual phrases. Another line of recent inquiry are dialog based image search systems [22, 10]. Liao et al [22] proposed to aggregate multi-round user responses from trained agents or human agents in order to iteratively refine a retrieved set of images using a hierarchical recurrent encoder-decoder framework [29]. We follow a similar protocol, but we explore a more open-ended domain of images corresponding to scenes depicting multiple objects. The method Guo et al [10] as in our work, used multiple rounds of natural language queries, and proposed collecting relative image captions as supervision for a product search task. In contrast, we pursue a weakly supervised approach where we leverage an image dataset with region captions that are used to simulate queries during training, thus bypassing the need to collect extra annotations. We demonstrate that training with simulated queries is surprisingly effective under human evaluations. As the hierarchical recurrent framework [29] was used in most of the previous dialog based methods [6, 7, 31, 22, 10], we provide a re-implementation of the hierarchical encoder (HRE) model with the queries as context and use it as one of our baselines. Different from the previous dialog based methods where the systems also provide textual responses, we explore a scenario where the system only responses with retrieved images, so no decoder module is required in our case.

Also relevant to our research are the existing works on learning image-word [14, 11, 21] or region-phrase [25] alignments for vision-language tasks. For instance, Karpathy et al [14] proposed to learn a bidirectional image-sentence mapping by jointly embedding fragments of images (objects) and sentences. The image fragments are extracted using a pre-trained object detector, while the sentence fragments are obtained using a dependency tree relation parser. Niu et al [25] extended this work by jointly learning hierarchical relations between phrases and image regions in an iterative refinement framework. Recently, Lee et al [21] developed a stacked cross attention network for word-region matching. Compared to these models, our proposed query state encoding aims at integrating multiple round queries while still using a compact representation of fixed size (i.e. independent of the number of queries), so that retrieval times do not depend on the number or the length of the queries. We show our compact representation to be both efficient and effective for interactive image search.

More closely related to our work are Memory Networks [35, 32, 15], which perform `query` and possibly `update` operations on a predefined memory space. In contrast to this line of research, we explore a more challenging scenario where the model needs to `create` and `update` the memory (i.e. the state vectors) on-the-fly so as to maintain the states of the queries.

## 3 Model

Retrieving images with multi-round refinements offers the potential benefit of reducing the ambiguity of each query but also raises challenges on how to integrate user queries from multiple rounds. Our model is inspired by the observation that users naturally underspecify in their queries by referring to local regions of the target image. We aim to capture these region level alignments by learning to map text queries $\{\mathbf{s}_t\}_{t=1}^{T}$ and the target image $\mathbf{I}$ into two sets of latent vectors $\{\mathbf{x}_i\}_{i=1}^{M}$ and $\{\mathbf{v}_j\}_{j=1}^{N}$ respectively, and computing the matching score of $\{\mathbf{s}_t\}_{t=1}^{T}$ and $\mathbf{I}$ by measuring and aggregating fine-grained similarities between $\{\mathbf{x}_i\}_{i=1}^{M}$ and $\{\mathbf{v}_j\}_{j=1}^{N}$. Figure 2 provides an overview of our model.

### 3.1 Image representation

To identify candidate regions referred in the queries, we follow [1, 21]. For each image $\mathbf{I}$, we first detect the potential objects and salient stuff using the FasterRCNN detector [27]. Corresponding

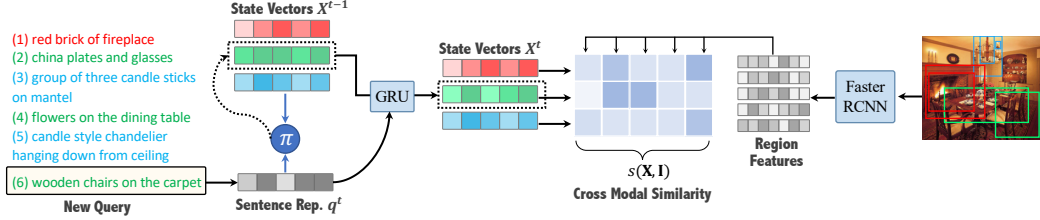

Figure 2: Overview of our model. Drill-down maintains a fixed set of state vectors $\mathbf{X}$, modeling the historical context of the user queries. Given a new query $\mathbf{q}^t$, our model selects and updates one of the state vectors. The updated state vectors $\mathbf{X}^t$ and image region features are then projected to a cross-modal embedding space to measure the fine-grained alignment between each region-state pair.

features $\{\mathbf{c}_j\}$ are extracted from the ROI pooling layer of the detector. In practice, we leverage the object detector provided by [1], which is pre-trained on Visual Genome [20] with 1600 predefined object and stuff classes. A linear projection $\mathbf{v}_j = W_I \mathbf{c}_j + b_I$ is applied to reduce $\{\mathbf{c}_j\}$ into D-dimensional latent vectors $\mathbf{V} = \{\mathbf{v}_j\}_{j=1}^N, \mathbf{v}_j \in \mathbb{R}^D$. Here $N$ is the number of regions in each image. The learnable parameters for the image representation $\{W_I, b_I\}$ are denoted as $\theta_I$.

## 3.2 Query representation

Supporting multi-round retrieval requires a state representation for integrating the queries from multiple turns. Solutions adopted by existing methods include applying a single recurrent network to the concatenation of all queries [9] or a hierarchical recurrent network [7, 31, 22, 10] modeling individual query and historical context in separate recurrent modules. These approaches produce a single latent vector which aggregates all queries. While state-of-the-art models [22, 10] show remarkable performance on domains such as fashion product search, we demonstrate that currently used single-vector representations are not the most effective for capturing complex scenes with multiple objects. Specifically, as image features used in existing methods are typically extracted from the penultimate layer of a pre-trained image classification or object detection model, input instances of the same or very similar categories activate the same feature units in the extracted feature space. Therefore, it is nontrivial for these latent representations to encode and distinguish multiple entities from the same or very similar categories (i.e. multiple `person` instances).

We propose to maintain a set of latent representations $\mathbf{X} = \{\mathbf{x}_i\}_{i=1}^M, \mathbf{x}_i \in \mathbb{R}^D$ for multiple turn queries. Here $M$ is the number of latent vectors. This parameter represents the computational budget, since retrieval time will depend on the compactness of this representation. While users might provide a general image description in the first round of querying, subsequent queries typically describe more specific regions. We aim at finding a good alignment between queries and image region representations $\{\mathbf{v}_j\}_{j=1}^N$. An ideal set of $\{\mathbf{x}_i\}_{i=1}^M$ should learn to group and encode the input queries into visually discriminative representations referring to distinct image regions. In the remaining of the section, we first introduce the cross modal similarity formula used in our model. We then explain how to update the state representations $\{\mathbf{x}_i\}_{i=1}^M$ from the queries $\{\mathbf{s}_t\}_{t=1}^T$ so as to optimize their matching score with the target image.

## 3.3 Cross modal similarity

To measure the similarity of $\mathbf{X} = \{\mathbf{x}_i\}_{i=1}^M$ and $\mathbf{V} = \{\mathbf{v}_j\}_{j=1}^N$, we first compute the cosine similarity of each possible state-region pair $(\mathbf{x}_i, \mathbf{v}_j)$: $s(\mathbf{x}_i, \mathbf{v}_j) = \mathbf{x}_i^T \mathbf{v}_j / \|\mathbf{x}_i\| \|\mathbf{v}_j\|$, where $\|.\|$ denotes the $L2$ norm. Given $s(\mathbf{x}_i, \mathbf{v}_j)$, we define the similarity $s(\mathbf{x}_i, \mathbf{I})$ between a state vector $\mathbf{x}_i$ and the target image $\mathbf{I}$ as

$$s(\mathbf{x}_i, \mathbf{I}) = \frac{1}{N} \sum_{k=1}^N \alpha_{ik} s(\mathbf{x}_i, \mathbf{v}_k), \quad \alpha_{ik} = \frac{\exp(s(\mathbf{x}_i, \mathbf{v}_k)/\sigma)}{\sum_j^N \exp(s(\mathbf{x}_i, \mathbf{v}_j)/\sigma)} \tag{1}$$

Here $\sigma$ is a temperature hyper-parameter. Note that this formulation is similar to measuring the cosine similarity of $\mathbf{x}_i$ and a context vector $\sum_{k=1}^{N} \alpha_{ik}\mathbf{v}_k$ from an attention module [24, 21]. The cross modal similarity between the state vectors $\mathbf{X} = \{\mathbf{x}_i\}_{i=1}^{M}$ and the target image $\mathbf{I}$ is defined as $s(\mathbf{X}, \mathbf{I}) = \frac{1}{M}\sum_{k=1}^{M} s(\mathbf{x}_k, \mathbf{I})$.

## 3.4 Query encoding

Given a query input $\mathbf{s}_t$ at time t, our model maps each word token $\mathbf{w}_k$ in $\mathbf{s}_t$ to an E-dimensional vector via a linear projection: $\mathbf{e}_k = W_E\mathbf{w}_k$, $\mathbf{e}_k \in \mathbb{R}^E$, $k = 1, \cdots, K$, then generates the sentence embedding via a uni-directional recurrent network $\phi$ with gated recurrent units (GRU) as: $\mathbf{h}_k = \phi(\mathbf{e}_k, \mathbf{h}_{k-1})$, $\mathbf{h}_k \in \mathbb{R}^D$. The first hidden state of $\phi$ is initialized as a zero vector, while the last hidden state is treated as the sentence representation: $\mathbf{q}^t = \mathbf{h}_K$. We also explore using a bidirectional encoder but find no improvement. Given the assumption that each text query describes a sub-region of the image, each $\mathbf{q}^t$ only updates a subset of the state vectors. In this work, we focus on a simplified scenario where each $\mathbf{q}^t$ only updates a single state vector $\mathbf{x}_k^{t-1} \in \mathbf{X}^{t-1}$. In detail, given the text query $\mathbf{q}^t$ at time step $t$, our model samples $\mathbf{x}_k^{t-1}$ from the previous state vector set $\mathbf{X}^{t-1} = \{\mathbf{x}_i^{t-1}\}_{i=1}^{M}$ based on the probability:

$$\pi(\mathbf{x}_k^{t-1}|\mathbf{X}^{t-1}, \mathbf{q}^t) = \begin{cases} \dfrac{\mathbb{1}(\mathbf{x}_k^{t-1}=\emptyset)}{\sum_j \mathbb{1}(\mathbf{x}_j^{t-1}=\emptyset)} & \text{if } \mathbf{X}^{t-1} \text{ has an } \textit{empty vector} \\[2em] \dfrac{\exp(f(\mathbf{x}_k^{t-1},\mathbf{q}^t))}{\sum_j \exp(f(\mathbf{x}_j^{t-1},\mathbf{q}^t))} & \textit{otherwise} \end{cases} \tag{2}$$

$$f(\mathbf{x}_k^{t-1}, \mathbf{q}^t) = W_\pi^3(\delta(W_\pi^2(\delta(W_\pi^1[\mathbf{x}_k^{t-1}; \mathbf{q}^t] + b_\pi^1)) + b_\pi^2)) + b_\pi^3, \tag{3}$$

where $\mathbb{1}(\mathbf{x}_j^{t-1} = \emptyset)$ is an indicator function which returns 1 if $\mathbf{x}_j^{t-1}$ is an empty vector and 0 otherwise. $f(\cdot)$ is a multilayer perceptron mapping the concatenation of $\mathbf{x}_k^{t-1}$ and $\mathbf{q}^t$ into a scalar value. Here $\delta$ is the ReLU activation function, $W_\pi^1 \in \mathbb{R}^{D\times 2D}$, $W_\pi^2, \in \mathbb{R}^{D\times D}$, $W_\pi^3 \in \mathbb{R}^{1\times D}$, $b_\pi^1, b_\pi^2 \in \mathbb{R}^D$, $b_\pi^3 \in \mathbb{R}$ are model parameters. An empty state vector is initialized with zero values. Ideally, an expressive sample policy should learn to allocate a new state vector when necessary. However, we empirically find it beneficial to update $\mathbf{q}^t$ to an empty state vector whenever possible. Once $\mathbf{x}_k^{t-1}$ is sampled, we update this state vector using a single uni-directional gated recurrent unit cell (GRU Cell) $\tau$: $\mathbf{x}_k^t = \tau(\mathbf{q}^t, \mathbf{x}_k^{t-1})$. Note that our formulation is similar to a hard attention module [37]. Leveraging a soft attention is possible, but it is more computationally expensive as it would need to update all state vectors. Our state vector update mechanism is inspired by the knowledge base methods with external memory [22]. Our method can be interpreted as building a knowledge base memory online from scratch, only from the query context, which can be trained end-to-end with other modules. We denote the learnable parameters for the state vector update policy function $\pi(\cdot)$ as $\theta_\pi = \{W_\pi^1, W_\pi^2, W_\pi^3, b_\pi^1, b_\pi^2, b_\pi^3\}$, and for the rest modules as $\theta_q = \{W_E, \phi, \tau\}$.

## 3.5 End-to-end training

Our model is trained to optimize $\theta_I$, $\theta_\pi$ and $\theta_q$ so as to achieve high similarity score between the queries $\{\mathbf{s}_t\}_{t=1}^{T}$ and the target image $\mathbf{I}$. Thus, we follow [9, 21] and adopt a triplet loss on $s(\mathbf{X}, \mathbf{I})$ with hard negatives:

$$L_e = \operatorname*{argmin}_{\theta_I, \theta_q} \sum_{\mathbf{X}, \mathbf{I}} \ell(\mathbf{X}, \mathbf{I})$$
$$\ell(\mathbf{X}, \mathbf{I}) = \max_{\mathbf{I}'}[\alpha + s(\mathbf{X}, \mathbf{I}') - s(\mathbf{X}, \mathbf{I})]_+ + \max_{\mathbf{X}'}[\alpha + s(\mathbf{X}', \mathbf{I}) - s(\mathbf{X}, \mathbf{I})]_+ \tag{4}$$

Here, $\alpha$ is a margin parameter, $[\cdot]_+ \equiv \max(\cdot, 0)$. $\mathbf{I}'$ and $\mathbf{X}'$ are decoy images and state vectors within the same mini-batch as the ground-truth pair $(\mathbf{X}, \mathbf{I})$ during training. Note that $L_e$ will only optimize the parameters $\theta_I$ and $\theta_q$. Directly optimizing $\theta_\pi$ is difficult as sampling from Equation 2 is non-differentiable. We propose to train the policy parameters via Reinforcement Learning (RL).

Formally, the state in our RL formulation is the set of state vectors $\mathbf{X}^t = \{\mathbf{x}_i^t\}_{i=1}^M$, and the action $k \in \{1, ..., M\}$ is to select the state vector $\mathbf{x}_k^t$ from $\mathbf{X}^t$ when fusing information from the embedded query vector $\mathbf{q}^{t+1}$. The RL objective is to maximize the expected cumulative discounted rewards, so in our case we define the reward function as the similarity between the state vectors $\mathbf{X}^t$ and the image $\mathbf{I}$, i.e. $s(\mathbf{X}^t, \mathbf{I})$. Note that our reward function evaluates the potential similarity at all future time step instead of only the last step $T$, encouraging the model to find the target image with fewer turns.

**Supervised pre-training**    As optimizing the sampling policy requires reward signals from the retrieval environment, we pre-train the model by optimizing $L_e$ with a fixed policy: $\pi(\mathbf{x}_k^{t-1} | \mathbf{X}^{t-1}, \mathbf{q}^t) = \mathbb{1}(k \equiv t \ (\text{mod M}))$, where $\mathbb{1}(\cdot)$ is an indicator function and $M$ is the number of state vectors. Intuitively, this policy circularly updates the state vectors in order.

**Joint optimization**    Given the pre-trained environment, we then jointly optimize the sampling policy and the other modules (i.e. $\theta_I$, $\theta_q$ and $\theta_\pi$). Because the next state $\mathbf{X}^{t+1}$ is a deterministic function given the current state $\mathbf{X}^t$ and action $k$, we adopt the policy improvement strategy from [10] to update the policy. Specifically, we estimate the state-action value $Q(\mathbf{X}^t, k) = \sum_{t'=t}^{T-1} \gamma^{t'-t} s(\mathbf{X}^{t'+1}, \mathbf{I})$ for each state vector selection action $k$ by sampling one look-ahead trajectory. $\gamma$ is the discount factor. The policy is then optimized to predict the most rewarding action $k^* = \text{argmax}_k Q(\mathbf{X}^t, k)$ via a cross entropy loss:

$$L_\pi = \underset{\theta_\pi}{\text{argmin}} \sum_{\mathbf{X}^t, \mathbf{q}^{t+1}} - \log(\pi(\mathbf{x}_{k^*}^t | \mathbf{X}^t, \mathbf{q}^{t+1}; \theta_\pi)) \tag{5}$$

We also jointly finetune $\theta_I$ and $\theta_q$ by applying $L_e$ on the rollout state vectors $\mathbf{X}_*$: $L_e^* = \text{argmin}_{\theta_I, \theta_q} \sum_{\mathbf{X}_*, \mathbf{I}} \ell(\mathbf{X}_*, \mathbf{I})$. The model is trained with the multi-task loss: $L = L_e^* + \mu L_\pi$, where $\mu$ is a scalar factor determining the trade-off between the two terms.

## 4   Experiments

**Dataset**    We evaluate the performance of our method on the Visual Genome dataset [20]. Each image in Visual Genome is annotated with multiple region captions. We preprocess the data by removing duplicate region captions (e.g. multiple captions that are exactly the same), and images with less than 10 region captions. This preprocessing results in 105,414 image samples, which are further split into 92,105/5,000/9,896 for training/validation/testing. We also ensure that the images in the test split are not used for the training of the object detector [1]. All the evaluations, including the human subject study, are performed on the test split, which contains 9,896 images. We use region captions as queries to train our model, thus bypassing the challenging issue of data collection for this task. The vocabulary of the queries is built with the words that appear more than 10 times in all region captions, resulting in a vocabulary size of 14,284. During training, queries and their orders are randomly sampled. During validation and testing, the queries and their orders are kept fixed.

**Baselines**    We compare our method with four baseline models: (1) **HRE**: a hierarchical recurrent encoder network, which is commonly adopted by recent dialog based approaches [31, 22, 10]. We consider the framework using text queries as context, which consists of a sentence encoder, a context encoder and an image encoder. The sentence encoder has the same word embedding (e.g. the linear projection $W_E$) and sentence embedding (e.g. the $\phi$ function) as the proposed model. The context encoder is a uni-directional GRU network $\psi$ that sequentially integrates the sentence features $\mathbf{q}^t$ from $\phi$ and generates the final query feature $\bar{\mathbf{x}}^t$ : $\bar{\mathbf{x}}^t = \psi(\mathbf{q}^t, \bar{\mathbf{x}}^{t-1})$. $\bar{\mathbf{x}}^0$ is initialized as a zero vector. The image encoder maps the mean-pooled features of ResNet152 [13] into a one-dimensional feature vector $\bar{\mathbf{v}}$ via a linear projection. The ResNet model is pre-trained on ImageNet [8]. The model is trained to optimize the cosine similarity between $\bar{\mathbf{x}}^t$ and $\bar{\mathbf{v}}$ by a triplet loss with hard negatives as in [9]. (2) **R-HRE**: a model similar to baseline (1) but is trained with the region features $\{\mathbf{v}_j\}_{j=1}^N$, as in the proposed method. Specifically, the model learns to optimize the similarity term $s(\bar{\mathbf{x}}^t, \mathbf{I})$ defined in Eq.(1) by a triplet loss with hard negatives similar to $L_e$ on one state vector. (3) **R-RE**: a model similar to baseline (2) but instead of using a hierarchical text encoder, this baseline uses a single uni-directional GRU network which encodes the concatenation of the queries. (4) **R-RankFusion**: a

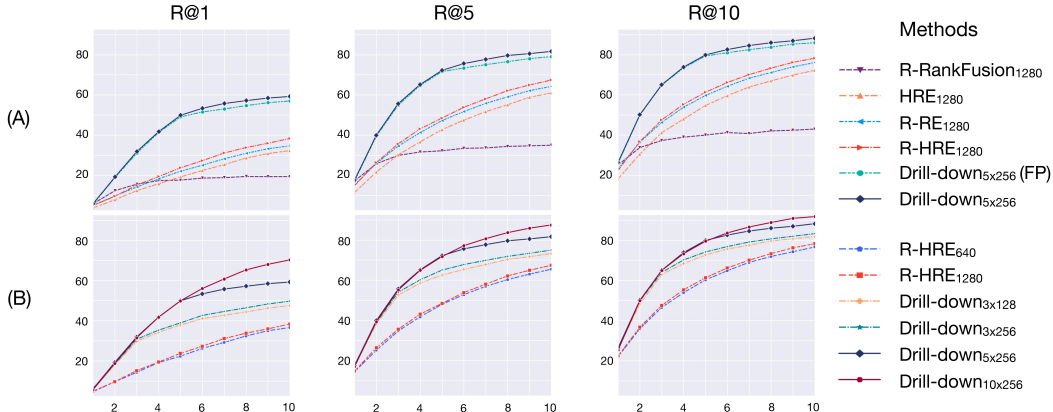

Figure 3: Quantitative evaluation of our models and the baselines. (A) Comparison of models using query representations of the same memory size; (B) Comparison of the models using query representations of different memory sizes. The horizontal axis represents the query turn.

| Methods | HRE/R-RE$_{1280}$ | R-HRE$_{640/1280}$ | Drill-down$_{3\times128}$ / $_{3\times256}$ / $_{5\times256}$ / $_{10\times256}$ |
|---|---|---|---|
| # Query Rep. | 1280 | 640 / 1280 | 384 / 768 / 1280 / 2560 |
| # Image Rep. | 1280 / 36 × 1280 | 36×640 / 36 × 1280 | 36 × 128 / 36 × 256 / 36 × 256 / 36 × 256 |
| # Parameters | 22820k | 9866k / 22820k | 4861k / 5830k / 5830k / 5830k |

Table 1: Sizes of the query/image representations and the parameters in our models and the baselines.

model where each query is encoded by a uni-directional GRU network and each image is represented as a set of region features $\{\mathbf{v}_j\}_{j=1}^N$. The ranks of all images are computed separably for each turn. The final ranks of the images are represented as the averages of the per-turn ranks.

**Implementation details**  We try to keep consistent configurations for all the models in our experiments to better evaluate the contribution of each component. In particular, all the models are trained with 10-turn queries ($T = 10$). We use ten turns as we'd like to track and demonstrate the performance of all methods in both short-term and long-term scenarios. For each image, we extract the top 36 regions ($N = 36$) detected by a pretrained Faster RCNN model, following [1]. Each embeded word vector has a dimension of 300 ($E = 300$). In all our experiments, we set the temperature parameter $\sigma$ to 9, the margin parameter $\alpha$ to 0.2, the discount factor $\gamma$ to 1.0, and the trade-off factor $\mu$ to 0.1. For optimization, we use Adam [16] with an initial learning rate of $2e - 4$ and a batch size of 128. We clip the gradients in the back-propagation such that the norm of the gradients is not larger than 10. All models are trained with at most 300 epochs, validated after each epoch. The models which perform best on the validation set are used for evaluation.

**Evaluation metrics**  To measure the retrieval performance, we use the common R@K metric, i.e., recall at K - the ratio of queries for which the target image is among the top-K retrieved images. The R@1, R@5 and R@10 scores at each turn are reported as shown in Fig. 3.

## 4.1  Results on simulated user queries

Due to the lack of existing benchmarks for multiple turn image retrieval, we use the annotated region captions in Visual Genome to mimic the user queries. As region captions focus more on invariant information, such as image contents, and convey fewer irrelevant signals, such as different speaking/writing styles, they could be seen as the common "abstracts" of real queries in different forms. While we agree that strong supervisory signals such as real user queries could bridge the domain gap and would like to explore further in this direction, we choose at this stage to use only "weak but free" signals and investigate their potentials of being generalized to real scenarios. First, we compare our method against the baseline models when using query representations of the same memory size. In particular, we use 5 state vectors in our model ($M = 5$), each with a dimension of 256. Accordingly, the baseline models use a 1280-d query vector. Figure 3(A) shows the per-turn

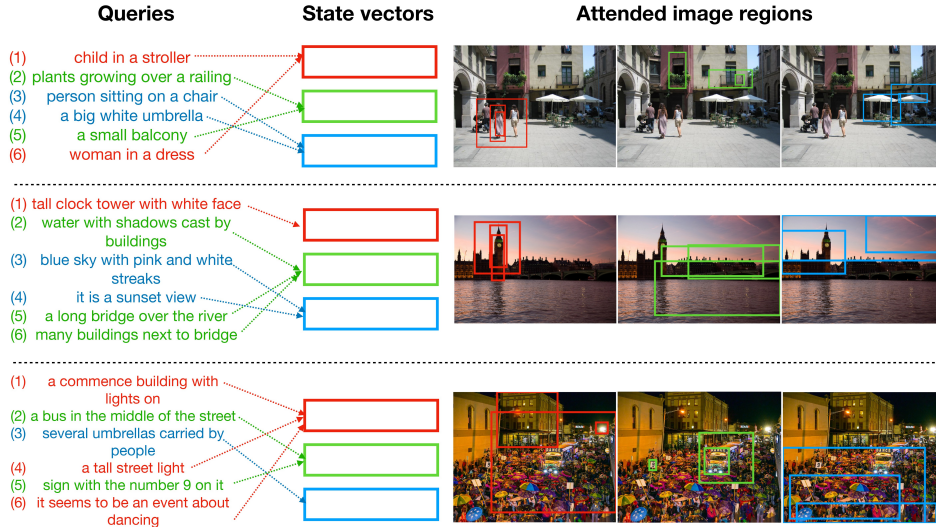

| Queries | State vectors | Attended image regions |
|---|---|---|

(1) child in a stroller
(2) plants growing over a railing
(3) person sitting on a chair
(4) a big white umbrella
(5) a small balcony
(6) woman in a dress

(1) tall clock tower with white face
(2) water with shadows cast by buildings
(3) blue sky with pink and white streaks
(4) it is a sunset view
(5) a long bridge over the river
(6) many buildings next to bridge

(1) a commence building with lights on
(2) a bus in the middle of the street
(3) several umbrellas carried by people
(4) a tall street light
(5) sign with the number 9 on it
(6) it seems to be an event about dancing

Figure 4: Qualitative examples of Drill-down$_{3\times128}$. The sequential queries and the corresponding state vectors used to integrate them are shown on the left; The top-3 regions of the target images attended by each state vector are shown on the right, with the same color as the corresponding state vector. Note that all these target images rank top-1 given the input queries.

performance of the models on the test set. Here Drill-down$_{5\times256}$(FP) indicates the supervised pre-trained model with the fixed policy, and Drill-down$_{5\times256}$ indicates the jointly optimized model with a learned policy. Both the R-RE$_{1280}$ and R-HRE$_{1280}$ baselines perform better than the HRE$_{1280}$ model, demonstrating the benefit of incorporating region features. R-HRE$_{1280}$ is superior to R-RE$_{1280}$, demonstrating the benefit of hierarchical context encoding. R-RankFusion$_{1280}$ performs inferior to all other models. Note that it also requires more memory to store the ranks of all images at each turn. Our models significantly outperform all baselines by a large margin. On the other hand, we observe that the performance of our model will degrade when different queries have to share the same state vector. For example, after the 5th turn, the Drill-down$_{5\times256}$(FP) model gains less improvement from each new query. Drill-down$_{5\times256}$ further improves Drill-down$_{5\times256}$(FP) by learning to distribute the queries into the most rewarding state vectors.

To investigate the design space of the query representation, we further explore variants of our model with different numbers of state vectors and feature dimensions. Table 1 shows the sizes of the query/image representations and the parameters used in our models and the baselines. Note that the R-RankFusion and R-RE models have the same size of query/image representations and parameters. Here Drill-down$_{M\times D}$ indicates the model with M state vectors, each with a dimension of D. As shown in Figure 3(B), while both Drill-down and the R-HRE baseline can be improved by increasing the feature dimension, using more state vectors gains significantly more improvements with the same, or even less memory budget. For example, Drill-down$_{3\times128}$ significantly outperforms R-HRE$_{1280}$ with 3 times less query features, 10 times less region features and 4 times less parameters. The highest performance is achieved by the model which stores each query in a distinct state vector: 10 state vectors for 10-turn queries. Integrating multiple queries into the same state vector could make the model "forget" the responses from earlier turns, especially when they activate the same semantic space as the new query.

Figure 4 provides qualitative examples of the Drill-down$_{3\times128}$ model. Here the arrows indicate the predicted state vectors used to incorporate the queries. We show the top-3 regions of the target images that have the highest similarity scores with each state vector (illustrated with the same color). We observe that the model tends to group queries with entities that potentially coincide with each other. However, it could also lead to the "forgetting" of earlier queries. For instance, in the first example, when aggregating the queries "*child in a stroller*" and "*woman in a dress*" in order, the model tends to focus on "*woman*" while forgetting information about "*child*", as "*woman*" and "*child*" potentially activate the same semantic subspace.

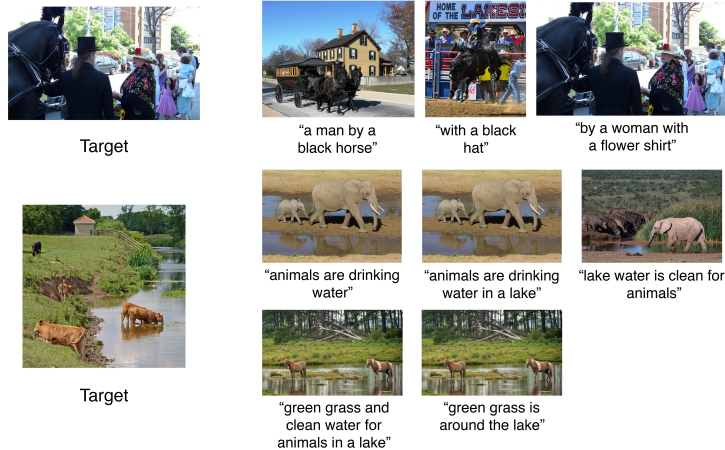

| | | | |
|---|---|---|---|
| Target | "a man by a black horse" | "with a black hat" | "by a woman with a flower shirt" |
| Target | "animals are drinking water" | "animals are drinking water in a lake" | "lake water is clean for animals" |
| | "green grass and clean water for animals in a lake" | "green grass is around the lake" | |

Figure 5: Examples of real user queries and the top-1 images from Drill-down$_{3\times256}$.

## 4.2 Results on real user queries

We evaluate our method with the queries from crowdsourced human users via a multi-round interactive system adapted from [3]. Given a target image, a user is asked to search for it by providing descriptions of the image content. The system shows top-5 retrieved images to the user per turn as context so that the user can improve the results by providing additional descriptions. This process is repeated until the image is found or it reaches 5 turns. We sample 80 random images from the test set and evaluate HRED$_{1280}$, R-HRED$_{1280}$ and Drill-down$_{3\times256}$ on these images respectively. Each image is viewed by 3 different users. For each model, the best result on each image is selected across users to ensure high quality responses. As shown in Figure 6, most users ($> 80\%$) successfully find the target image within 5 turns, demonstrating the effectiveness of the multi-round search paradigm and the quality of using region captions for training. In particular, Drill-down$_{3\times256}$ consistently outperforms HRE$_{1280}$ and R-HRE$_{1280}$ on all evaluation metrics. On the other hand, as real user queries have more flexible forms, e.g. longer sentences, repeated descriptions of the same region, etc, we also observe smaller performance gaps between our method and the baselines. We believe further efforts such as real query data collection are needed to systematically fill this domain gap. Figure 5 shows example real user queries and the retrieval sequences using Drill-down$_{3\times256}$.

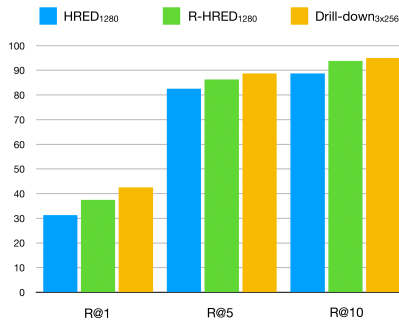

Figure 6: Human subject evaluation of the HRE$_{1280}$, R-HRE$_{1280}$ baselines and our Drill-down$_{3\times256}$ model.

## 5 Conclusion

We present Drill-down, a framework that is efficient and effective in interactive retrieval of specific images of complex scenes. Our method explores in depth and addresses several challenges in multiple round retrievals with natural language queries such as the compactness of query state representations, and the need for region-aware features. It also demonstrates the effectiveness of training a retrieval model with region captions as queries for interactive image search under human evaluations.

**Acknowledgements** We thank our anonymous reviewers for helpful feedback. This work was funded by a research grant from SAP Research and generous gift funding from SAP Research. We thank Tassilo Klein and Moin Nabi from SAP Research for their support.

## Footnotes

[1]Codes are available at `https://github.com/uvavision/DrillDown`

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
