[Supplementary Material]

# Examples of Real User Queries

Figure 1: Examples of real user queries collected in the human subject study and the top-3 retrieved images from the Drill-down$_{3\times256}$ model at each turn. The ranks of the target image (A) at each turn are 121, 32, 14, 1. The ranks of the target image (B) at each turn are 42, 9, 3, 1

T1: "a boat docked"

T2: "a docked white boat with blue trimming"

T3: "a docked white boat with blue trimming and a forest in the back"

T4: "a docked white boat with blue trimming and a forest in the back and a red boat in the background"

T5: "a docked white boat with blue trimming and a forest in the back and a red boat in the background and stones in the water"

Target

T1: "there are two women getting ready on opposite sides of a wall"

T2: "a woman on the left is wearing a white dress"

T3: "the image is of two rooms"

T4: "the image is black and white"

T5: "two women getting ready in separate rooms"

Target

(A)

(B)

Figure 2: Examples of real user queries collected in the human subject study and the top-3 retrieved images from the Drill-down$_{3\times256}$ model at each turn. The ranks of the target image (A) at each turn are 51, 24, 11, 7, 3. The ranks of the target image (B) at each turn are 182, 23, 7, 2, 1.

T1: "it is a bathroom"

T2: "there is a single sink on the left"

T3: "the toilet is in the middle and a tub on the right"

T4: "there is a dark vanity"

Target

(A)

T1: "search for humans in the back of a truck"

T2: "yellow truck with people in the back near a building"

T3: "men in the back of a truck turning right near a building with street lights to the right"

T4: "multiple men in the back of a beige truck"

Target

(B)

Figure 3: Examples of real user queries collected in the human subject study and the top-3 retrieved images from the Drill-down$_{3\times256}$ model at each turn. The ranks of the target image (A) at each turn are 33, 10, 10, 1. The ranks of the target image (B) at each turn are 826, 62, 24, 4.