[Reviews · NeurIPS 2019]

Reviewer 1



1. The main problem for me is that the paper promises a very real scenario (Fig. 1) of how a user can refine search by using a sequence of refined queries. However, majority of the model design and evaluation (except section 4.2) is performed with dense region captions that have almost no sequential nature. While this is partially a strength as no additional labels are required, the method seems suited especially towards such disconnected queries -- there is space for M disconnected queries and only then updates are required. 2. It would be good to have simple baselines that are modified suitably for multi-query retrieval. This would provide a deeper understanding of when the proposed method works better. For example, (a) scoring each query separately and performing late ranking fusion (either scores or ranks); (b) concatenating the query and performing retrieval through a typical joint text-image embedding. 3. Results on real user queries are most critical. In Fig. 1, the user queries seem very natural, but the simulated queries in Fig. 4 are not. It is not clear from the provided information, whether the users learn to create queries that just list the objects in the image, or have a more natural pattern of searching for the correct scene, followed by bigger objects/elements, and then their details. Overall: I'm not convinced of the paper in it's current state, although it is quite borderline. Personally, when doing image search, I noticed that I'm mostly searching for images with a single big object. It is unclear whether users need to find images like those shown in Fig. 4 (middle) and would not rather search using terms like "London skyline", and refine with "Big-Ben tower" instead of vague comments like "water with shadows cast by building". Additionally, Google image search provides a few options based on attributes such as clipart vs. drawing / black-and-white vs. specific color tones / etc. This crucial aspect seems to be missing in the current paper. ----------------------------- Post-rebuttal: The rebuttal does clarify some aspects about difference between user queries and dense captions (used during training). However, more examples and user queries would be required to fully ascertain that this is a viable option. There are a lot of todo experiments as well for the final version. I update my rating to reflect the score assuming these are incorporated into the final version.

Reviewer 2



The approach is an reasonable approach that allows a user to interactively "drill down" to find a desired image. The evaluation is fairly strong, including both simulated queries based on Visual Genome annotations and a user study that asks users to iteratively construct queries to focus in on a desired image, and compares to reasonable baselines representing the state of the art on dialog-based image retrieval The approach is new, but relatively simple and straightforward, using multiple user query encodings and a learned similarity that allows it to retrieve images that match all of these queries The language encoding uses uni-directional GRU. Why not bidirectional? Why not use a more recent transformer-based language encoding such as BERT? These have been shown to produce better language embeddings for most NL problems. The user experiments give users an image and then have them iteratively search for it in the corpus. Does this accurately model the real world problem where people are searching for an image that they are not actually looking at while they are constructing queries? It is a reasonable experimental methodology that allows effectively quantitative evaluation but I wonder if it truly models a realistic scenario.

Reviewer 3



*Originality* This paper introduces a new approach for language-based image retrieval in a multi-turn setting. The method is built on Visual Genome, using image region descriptions as placeholders for sentences in a multi-turn setting. The concept of updating an external memory has been explored extensively in the past (another relevant paper is Kiddon et al. 2016, neural checklist models). Dialogue in image tasks has been explored previously, so my understanding is the main contribution of this paper is a new way of encoding dialogue turns. Distinguishing the paper’s contribution with memory networks (with respect to the sentence encoding) would be important to add to the paper. Another related paper is Visual Dialog (Das et al. 2017). *Quality* Most experiments were conducted on Visual Genome, which was not built for this task. While some conclusions (such as learning capacity with different encoders) could perhaps be drawn using this data, conclusions about the general task or how humans would interact with a retrieval system are less supported by the experiments on Visual Genome. Claims about HCI should be cited (e.g., “users naturally underspecify in their queries by referring to local regions of the target image”, L104). Critical experimental details are missing (or I couldn’t find them). For example, (1) How is the order of region descriptions chosen during training or testing? (2) How is the subset of 10 region descriptions chosen? (3) Why are images with fewer than 10 region captions removed for training, but during human evaluation, dialogues can reach only up to 5 turns? (4) From what set of images are images retrieved -- the set of all image in the dataset, or just the testing set? Is it different during human evaluation? (5) How are images scored for retrieval -- is inference run over all images in the candidate set and the highest returned? Is the set of candidate images narrowed as more sentences are provided? (6) Are result in Figure 3 shown on the development or testing set? There seems to be a correlation between the number of placeholder vectors and performance (as shown in Figure 3). Accordingly, there are a few experiments that I would like to see: . Using the same number of placeholder vectors as sentences. (Based on how placeholders are sampled, this would be equivalent to applying a transformation on each sentence encoding). . Instead of using placeholder vectors, just directly computing attention with the sentence encodings and the image regions. Is learning the extra parameters for encoding into a placeholder necessary? I suppose it makes the attention computation more cost-effective for very long dialogues, but if dialogues range around 5-10 turns, then it doesn’t seem justified. . Similarly, the logic that always samples an empty placeholder while one exists seems unnecessary -- why not just allow the model to sample whichever placeholder it wants? It seems like sampling empty ones first could cause weird ordering effects (because the first N may not use the same placeholder). Some comments/questions about the human evaluation: . Why limit up to five turns? If evaluating with a 5 x 256 model, this just means each of their utterances is encoded in a different placeholder. . Did you evaluate the language used by the humans and how it differed from the Visual Genome data? I imagine it would be quite different because it’s highly conditioned on the distractors that appear in the first few queries. . How many queries are required (on average) before the target image is in the top N? . Human evaluation shouldn’t be done on the test set of the data -- this is a form of peeking. *Clarity* The paper was easy to read. Some details are missing (see above section), but the contribution is clear. Some small points: . L20: “We focus in” → “We focus on” . L45: “vector .” → “vector.” . L48--L50: This sentence should be split into two. . Is the FasterCNN updated during training? . The notation in Section 3.4 is confusing. \phi is commonly used for word embeddings, not RNNs. . L166: “computational” → “computationally” . Using two different notations for max in Eq 4 is confusing. . HRED stands for hierarchical recurrent encoder-decoder. This method doesn’t have a decoder. . Table 1 is very difficult to read with so many slashes. *Significance* The proposed method is easy to understand and could be applied to other tasks using sequential language understanding for vision (e.g., Visual Dialogue). However, using Visual Genome as a placeholder for the real task of image retrieval is not convincing (it was not annotated with this task in mind, and sampling a subset of region descriptions in no specific order means this method was not evaluated with ordering information in mind, as would be present in an image retrieval system). Also, important questions remain about the evaluation (missing baselines and the candidate set for retrieval). *Update after review* Thanks for including details about the inference and evaluation details in the rebuttal. These should be added to the main paper. After reading the rebuttal, my main concerns are that (1) Because the data wasn't created originally for a sequential language understanding task, it's hard to draw conclusions about the proposed method for such a task, though using human evaluation is a great way to do it. (2) Some choices about the architecture (e.g., learning placeholders rather than doing simple dot-product attention with encoded captions) should be motivated through more detailed experiments as described earlier in my review.

[Author Response · NeurIPS 2019]

We thanks reviewers for their feedback. First, we address some common concerns.

**Disparity between the region caption based queries in our experiments and the user queries in real scenario**.
As emphasized in our manuscript (L53), leveraging region captions for weak supervision can been seen as one of the advantages of our method, as no annotation is required. While we agree strong supervisory signals such as real user queries could bridge the domain gap and would like to explore further in this direction, we choose at this stage to use only "weak but free" signals to see to what extent they can be generalized to practical applications. We demonstrate the effectiveness of the proposed method on simulated data with extensive experiments, on real scenarios via human subject studies on Amazon Mechanical Turk. Our experiments show the region caption based queries can potentially be generalized to real scenarios with high promise.

**Patterns of real user queries**
Compared with the simulated caption data, the real user queries we collected show distinct patterns, e.g. long descriptions of the target scenes, less informative sentences (e.g. 'good!'). That's why we observe smaller performance gaps between our method and the alternative approaches in the user study. We will include more details of the user study and examples of the real user queries in the revised paper.

**Reviewer 1**
**#Q2**: **Baselines**
With relatively less prior works on this research, we compare the proposed method with variants of state-of-the-art approaches for the most related topics, e.g. dialog-based interactive product search. We will incorporate the suggested baselines (e.g. late score/rank fusion vs early feature fusion, linear query encoding vs hierarchical query encoding) in the revised paper.

**#Q4**: **Applications of the proposed method**
We envision the proposed method could generally help with natural image search. Potential applications include retrieving very specific images of complex scenes the users encountered before, or exploring inspiring images for creative content generation (e.g. Adobe Stock Image).

**Reviewer 2**
**#Q1**: **Incorporating advanced language models**
We have explored using bidirectional language encoders and found it performs similar with unidirectional encoders in this task. We conjecture that unidirectional and bidirectional encoders provide comparable contextual signals when encoding the per-turn query as a single feature vector for downstream modules. In the current manuscript, we focus more on the sequential encoding of multiple sentences, and would like to explore and incorporate more advanced language models such as BERT in the future.

**Reviewer 3**
**#Q1**: **Distinguishing the paper's contribution with memory networks**
In contrast to the previous sentence encoding methods which perform **query** and possibly **update** operations on a predefined external memory space (e.g. the agenda items in Kiddon et al. 2016, neural checklist models), we focus on a more challenging scenario where the model needs to **create** and **update** the memory module (the state vectors in our case) **on-the-fly** so as to maintain the dynamic states of multiple-turn queries. We will elaborate more to distinguish our method with memory networks in the revised paper.

**#Q2**: **Experimental details**
(1) The region captions and their orders are randomly sampled. We keep the captions and their orders of the validation and test sets unchanged for all our experiments; (2) We use ten turns in all our simulated experiments as we'd like to track and demonstrate the performance of the proposed method in both short-term and long-term scenarios, as shown in Fig. 3. In the user study, we start with ten-turn queries but observe the users are less willing to continue and finish the tasks if they could not succeed in five turns, so we evaluate the five-turn queries in our experiment; (3) We use different image sets for training, validation and evaluation (L208), where the images retrieved are from the corresponding sets at different stages respectively. All the evaluations (including the user study) are performed on the test set, which contains 9896 images (L208); (4) All images in the candidate set are ranked in each turn. (5) Faster RCNN is NOT finetuned in our experiments.

**#Q3**: **Number of the placeholders**
We confirm in our experiments that more placeholders result in better performance but also lead to scalability difficulties. One of the main goal of the proposed work is to model multiple-turn queries with dynamic lengths using a fixed set of hidden states (fixed computional budget accordingly). We're also happy to include the suggested experiments and improve the presentation of the paper in the revised version.

**#Q4**: **Questions about the human evaluation**
As answered in $\#Q2$, we start experimenting with ten queries but discover it cannot fit in the real scenario. We agree that using less placeholders will be more convincing in this case and will rerun the experiment in the revised paper.

[Meta-Review · NeurIPS 2019]

This paper investigates the problem of multi-round natural language image retrieval, using annotations from the Visual Genome dataset for training and evaluation. After feedback and reviewer discussion, this paper received final ratings of 6, 6 and 7. Despite some concerns about the use of non-sequential annotation data for a sequential task, the reviewers found the proposed model to be generally sound and the experimental evaluation convincing, and the AC agrees. However, we would encourage the authors to pay close attention to the reviewer feedback when preparing the final paper version. In particular, the author feedback committed to including the additional baselines requested by R1, so these should be included in the final version as promised.